# Generalised Implicit Neural Representations

**Daniele Grattarola**
EPFL
Lausanne, Switzerland
daniele.grattarola@epfl.ch

**Pierre Vandergheynst**
EPFL
Lausanne, Switzerland
pierre.vandergheynst@epfl.ch

## Abstract

We consider the problem of learning implicit neural representations (INRs) for signals on non-Euclidean domains. In the Euclidean case, INRs are trained on a discrete sampling of a signal over a regular lattice. Here, we assume that the continuous signal exists on some unknown topological space from which we sample a discrete graph. In the absence of a coordinate system to identify the sampled nodes, we propose approximating their location with a spectral embedding of the graph. This allows us to train INRs without knowing the underlying continuous domain, which is the case for most graph signals in nature, while also making the INRs independent of any choice of coordinate system. We show experiments with our method on various real-world signals on non-Euclidean domains.

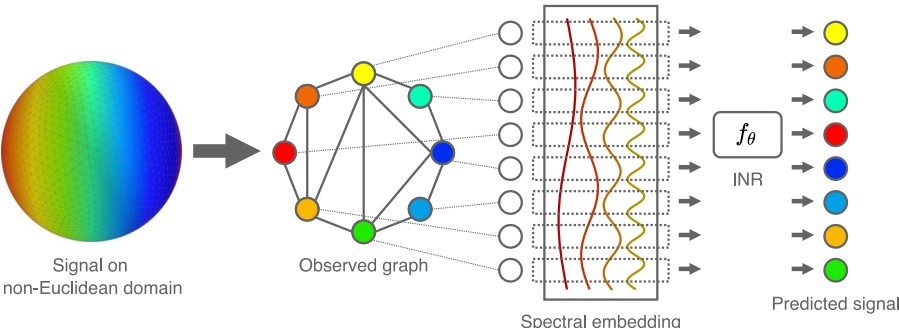

Figure 1: Given a continuous signal on a non-Euclidean domain, we observe a discrete graph realisation of it. A generalised implicit neural representation is a neural network $f_\theta$ trained to map a spectral embedding of each node to the corresponding signal value.

## 1 Introduction

Implicit neural representations (INRs) are a class of techniques to parametrise signals using neural networks [48, 41, 45, 34]. INRs are trained to map each point in a given domain to the corresponding value of a signal at that point. For example, INRs for images learn to map the 2D coordinates of pixels to their corresponding RGB values. INRs can also be conditioned on a latent vector, typically learned end-to-end as part of the model, that allows them to represent different signals on the same domain. INRs have been successfully applied to model complex signals like images [48, 45], signed distance functions [41], and radiance fields [40].

By expressing a signal as an INR, we obtain a continuous approximation of the signal on the whole domain. This allows us to compute a higher-resolution approximation of the signal by sampling more points on the domain (*e.g.*, a finer grid of pixels). Additionally, since INRs are fully differentiable,

36th Conference on Neural Information Processing Systems (NeurIPS 2022).

the latent vector of a conditional INR can be optimised with backpropagation and gradient descent to obtain a signal with some desired characteristics. For example, a conditional INR for signed distance functions can be used to design surfaces with a target aerodynamic profile [42].

So far, the literature on INRs has only focused on signals on Euclidean domains. However, signals defined on non-Euclidean domains are ubiquitous in nature and are especially relevant in artificial intelligence, as demonstrated by the recent rise of geometric deep learning [12]. In this paper, we propose an extension of the INR setting to signals on arbitrary non-Euclidean domains.

**Contributions**    We formulate the *generalised* INR problem as the task of learning an implicit representation for a signal on an arbitrary topological space $\mathcal{T}$ where, instead of observing the signal sampled on a regular lattice, we observe a graph (*i.e.*, a discretisation of $\mathcal{T}$) and the corresponding graph signal. In most practical cases, $\mathcal{T}$ is unknown and we cannot represent the sampled vertices in a coordinate system. Even when $\mathcal{T}$ is known (as in the Euclidean case), training an INR on a fixed coordinate system means that the model will depend on this choice. We solve both these issues by identifying the sampled nodes with an intrinsic spectral embedding obtained from the eigenvectors of the graph Laplacian. We then train a neural network to map the spectral embeddings to the corresponding signal values. Figure 1 shows a schematic view of the method.

Since the eigenvectors of the graph Laplacian are a discrete approximation of the continuous eigenfunctions of the Laplace-Beltrami operator on $\mathcal{T}$ (when appropriately rescaled) [5, 10, 7], at inference time we can map arbitrary points on $\mathcal{T}$ to the corresponding approximation of the signal, as long as the discrete graph signal is sampled consistently. This allows us to compute higher-resolution signals or to estimate the signal on different graph realisations of the same phenomenon (*e.g.*, different social networks with a similar structure).

**Results**    In the experiments section, we show concrete examples of learning INRs for signals on graphs and manifolds, using real-world data from biology and social networks. We also show that INRs trained on one instance of a graph can be transferred to different graph realisations of the underlying domain. Then, we show applications of conditional generalised INRs to model different signals on the same irregular domain, and also different signals on different domains. Finally, we conclude the paper with an experiment that consolidates all our results, modelling real-world meteorological signals on the spherical surface of the Earth.

## 2    Related works

INRs have recently attracted the attention of the machine learning community for their ability to represent complex signals, especially in computer vision, 3D rendering, and image synthesis applications [41, 38, 2, 22, 40, 46, 32]. Recent work has highlighted that INRs benefit from computing sinusoidal transformations of the coordinates, typically called Fourier features or positional encodings (PEs) [52, 50, 40, 53, 8]. Since the PEs typically used in INRs are also the eigenfunctions of the Laplace operator in Euclidean domains, our method is a generalisation of this approach.

However, these transformations are not applicable to non-Euclidean domains and still depend on a choice of coordinate system. A proposal for making INRs equivariant under the SO(3) group has been presented by Deng et al. [15], although this technique does not apply in general to non-Euclidean domains. The field of geometric deep learning has also explored the use of graph PEs similar to the ones we use in this paper, showing their usefulness as node features for graph neural networks [47, 36, 17, 16]. Recent research has also investigated the idea of learnable graph PEs [18, 13].

Related to our work, Belkin and Niyogi [5], Boguna et al. [11] and Levie et al. [30] have studied the geometrical link between graphs and topological spaces. Keriven et al. [25, 26] investigated the convergence, in the limit, of graph neural networks trained on large graphs. An alternative view of graphs as discrete samples of continuous spaces is the concept of graphon, which can be seen both as a generative model and as the limit of a sequence of graphs [37].

We also mention the work of Koestler et al. [28], who propose a similar idea to ours for learning fields on meshes. In this work, the authors train a neural network to map the eigenfunctions of the Laplacian to an RGB texture. By contrast, we focus on a wider range of tasks and experiments, including transferring to completely different graphs, conditioning the INRs, and modelling dynamical systems. Our scheme to compute generalised embeddings is also different from Koestler et al.'s, not relying on knowing node coordinates.

# 3   Method

**Standard setting**   In the standard INR setting, we consider a signal $f : \mathcal{X} \to \mathcal{Y}$ with $\mathcal{X} \subseteq \mathbb{R}^d$ and $\mathcal{Y} \subseteq \mathbb{R}^p$. We observe a discrete realisation of the signal $f(\mathbf{x}_i)$ for $i = 1, \dots, n$, where points $\mathbf{x}_i$ are sampled on a regular lattice on $\mathcal{X}$. Then, we train a neural network $f_\theta : \mathcal{X} \to \mathcal{Y}$, with parameters $\theta$, on input-output pairs $(\mathbf{x}_i, f(\mathbf{x}_i))$. Since we know that the signal domain is a subset of $\mathbb{R}^d$, at inference time we can sample points anywhere on $\mathcal{X}$ to compute the approximated value of the signal at those points. For example, an INR for images maps equispaced points in the unit square (pixel coordinates) to points in the unit cube (RGB values normalised between 0 and 1). The specific image on which we train the INR is a realisation of one such signal at a given resolution, and at inference time we can sample a finer lattice to super-resolve the image [48, 45].

**Generalised setting**   In the generalised setting, we consider a continuous signal $f : \mathcal{T} \to \mathcal{Y}$, with $\mathcal{T}$ an arbitrary topological space. We observe a discrete graph signal $f(v_i)$ on an undirected graph $G = (V, E)$, with node set $V = \{v_i\}$ for $i = 1, \dots, n$ and edge set $E \subseteq V \times V$. Note that the graph can be weighted if a metric is available on $\mathcal{T}$. The meaning of sampling $G$ from $\mathcal{T}$ is intuitive in the case of geometric meshes or other physical structures like proteins (in which case we also know the coordinates of $v_i$), but the same reasoning also applies to more complex domains with an abstract meaning (*e.g.*, the space of all possible papers and their citations). We generally assume that the graph describes some measure of closeness between uniformly sampled points on $\mathcal{T}$, be it a function of the coordinates (*e.g.*, a kernel) or some logical or functional relation that is given as part of the data. In general, we don't assume to know the true $\mathcal{T}$ from which $G$ is sampled or the coordinates of $v_i$. We only observe $G$ and the associated graph signal. For an in-depth discussion on the relation between graphs and topological spaces, see references [5], [11], and [30].

**Proposed method**   We approach the generalised INR problem by mapping $v_i$ to $f(v_i)$ through a spectral embedding obtained from the graph Laplacian. Let $\mathbf{A} \in \mathbb{R}^{n \times n}$ be the weighted adjacency matrix of graph $G$, $\mathbf{D}$ the diagonal degree matrix, and $\mathbf{L} = \mathbf{D} - \mathbf{A}$ the combinatorial Laplacian. The eigendecomposition of the Laplacian yields an orthonormal basis of eigenvectors $\{\mathbf{u}_k \in \mathbb{R}^n\}$ for $k = 1, \dots, n$, with a canonical ordering given by their associated eigenvalues $\lambda_1 \leq \lambda_2 \leq \cdots \leq \lambda_n$. We define a generalised spectral embedding of size $k$ for node $v_i$ as

$$\mathbf{e}_i = \sqrt{n} \left[ \mathbf{u}_{1,i}, \dots, \mathbf{u}_{k,i} \right]^\top \in \mathbb{R}^k. \tag{1}$$

The generalised spectral embeddings are a discrete approximation of the eigenfunctions of the continuous Laplace-Beltrami operator on $\mathcal{T}$ and converge to them for $n \to \infty$. See the discussion in references [5], [10], [9], and [7] for a formal analysis. In practice, rescaling the eigenvectors by $\sqrt{n}$ ensures that the embeddings for graphs of different sizes will have the same range component-wise and that similar nodes will have similar embeddings regardless of the graph size. As an example, we show a comparison between the eigenvectors and the generalised spectral embeddings of a path graph in Figure 2.

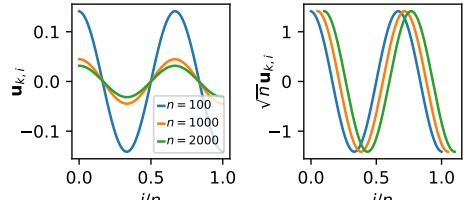

Figure 2: Laplacian eigenvectors (left) *vs.* rescaled spectral embeddings (right) of a path graph, for $k = 3$ and $n = 100, 1000, 2000$. The curves on the right are shifted to improve visualisation.

We train a neural network $f_\theta : \mathbf{e}_i \mapsto f(v_i)$ on the observed graph signal. Since the generalised spectral embeddings are an intrinsic property of the graph, the INR will not depend on any choice of coordinate system but only on the topology of the underlying continuous domain.

At inference time, we can compute the approximated value of the signal at an arbitrary location on $\mathcal{T}$ by computing the associated spectral embedding. If we know $\mathcal{T}$, or if we estimate it, we can sample new vertices directly from the domain and apply the same procedure used to construct the training graph. If $\mathcal{T}$ is unknown (*e.g.*, in the case of natural graphs that describe some abstract concept like citations, social interactions, or biological relations), then we just assume to observe a similar graph sampled from $\mathcal{T}$. An important difference between generalised INRs and Euclidean INRs is that generally we must observe the full graph in order to compute the inputs. This is necessary because, while in the Euclidean case we completely know the domain of the signal *a priori*, in the generalised case we need to estimate the topology of the domain by sampling.

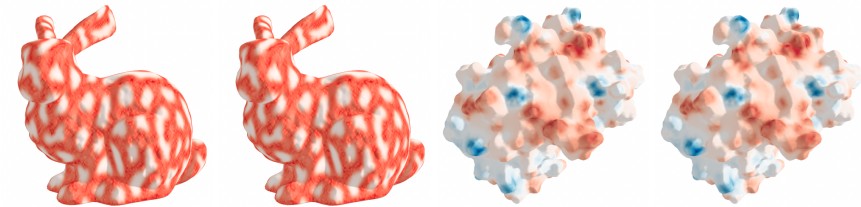

Figure 3: Ground truth signals *vs.* signals predicted by the INR. Best viewed in colour.

**Complexity**     Although computing the full eigendecomposition of the Laplacian has a complexity of $O(n^3)$, here we are only interested in the first $k$ eigenvectors. Also, the Laplacian of most graphs is very sparse, with nodes having an average degree $\bar{d} \ll n$. We can therefore use the implicitly restarted Lanczos method for eigendecomposition implemented by the ARPACK software, which has a complexity of $O(\bar{d}n^2)$ and can be easily parallelised [29]. In practice, all computations for this paper were easily managed on a commercial laptop with 10 CPU cores, scaling up to graphs in the order of $10^5$ nodes.

We also note that at inference time, depending on how the edges are constructed, it could be possible to add new nodes and edges to the training graph instead of sampling an entirely new graph from $\mathcal{T}$. In this case, the spectral embeddings for the new nodes can be estimated using the Nyström method without needing to compute the full eigendecomposition [3, 9].

**Alternative approaches**     The generalised Laplacian embeddings used in this paper are only one of many possibilities to represent nodes sampled from $\mathcal{T}$. To name a few, locally linear embeddings [43], Isomap [51], Laplacian eigenmaps [6] and diffusion maps [14] are all based on the idea of embedding data using the first few principal eigenvectors of a similarity matrix. Here we focus on the Laplacian since it is well known, sparse, and easy to compute, and its eigendecomposition is stable to graph perturbations. One disadvantage of Laplacian eigenvectors is that they are only unique up to sign, leading to $2^k$ possible eigenbases for a given $\mathcal{T}$ and the consequent ambiguity when transferring an INR to a different graph realisation. Additionally, eigenvalues of multiplicity greater than 1 also introduce ambiguity in the eigenvectors, since all rotations and reflections of the associated eigenspace are valid choices. However, simple heuristics can be used to eliminate sign ambiguity, and we did not encounter any other practical issue in this regard. Alternative ways to resolve the ambiguity is to use PEs based on random walks [18, 39, 31], heat kernels [49, 19], or the more recent sign-and-basis invariant neural networks [33]. We leave the exploration of these alternatives to future work, since they do not significantly impact our main contributions.

# 4   Experiments

**Setting**     We implement the generalised INR as a SIREN multi-layer perceptron [45]. The model has 6 layers with 512 hidden neurons and a skip connection from the input to the middle layer. We use the same hyperparameters and initialisation scheme suggested by Sitzmann et al. [45]. We train the model using Adam [27] with a learning rate of $10^{-4}$ and an annealing schedule that halves the learning rate if the loss does not improve for 1000 steps. At each step, we sample 5000 nodes randomly from the graph as a mini-batch. We use spectral embeddings of size $k = 100$, a value that we found empirically as a good trade-off between complexity and performance. We report an analysis on the effect of $k$ in Section 4.1. We ran all experiments on an Nvidia Tesla V100 GPU. The code to reproduce our results and the high-resolution version of all figures are available at `https://github.com/danielegrattarola/GINR`. All deviations from the default setting are documented in the supplementary material, where we also report an additional experiment on solving differential equations with generalised INRs.

## 4.1   Learning generalised INRs

We begin by verifying that a typical INR architecture can successfully learn arbitrary graph signals in the generalised setting. We consider three datasets as test cases.

**Bunny** We generate a texture on the Stanford bunny mesh[1] using the Gray-Scott reaction-diffusion model given by:

$$\Delta\mathbf{a} = -D_a\mathbf{L}\mathbf{a} - \mathbf{a}\odot\mathbf{b}\odot\mathbf{b} + F(\mathbf{1}_n - \mathbf{a}); \ \ \Delta\mathbf{b} = -D_b\mathbf{L}\mathbf{b} + \mathbf{a}\odot\mathbf{b}\odot\mathbf{b} - (F + K)\mathbf{b} \tag{2}$$

for state vectors $\mathbf{a}, \mathbf{b} \in \mathbb{R}^n$ and parameters $D_a, D_b, F, K \in \mathbb{R}$, where $\odot$ indicates element-wise product [21]. The mesh has 34834 nodes and 104288 edges. We configure the system to yield "coral" patterns[2] and evolve the simulation for $10^4$ steps starting from random positive values for $\mathbf{a}, \mathbf{b}$. At the end of the evolution, we use vector $\mathbf{a}$ as target graph signal.

**Protein** As a real-world domain, we consider the solvent excluded surface of a protein structure.[3] The continuous signal is the value of the electrostatic field generated by the amino acid residues at the surface. Protein function can be largely understood by studying their surfaces and the corresponding chemical features, like

Table 1: $R^2$ and mean squared error for the bunny, protein, and US election signals.

|  | **Bunny** | **Protein** | **US Election** |
|---|---|---|---|
| $R^2$ | 1.000 | 1.000 | 0.999 |
| MSE | $9.14 \cdot 10^{-8}$ | $1.17 \cdot 10^{-10}$ | $1.45 \cdot 10^{-3}$ |

the electrostatic charge, so this represents a potentially interesting application of INRs to biology. We use a combination of the MSMS [44] and APBS [24] software packages to estimate the surface, sample the nodes, and get the corresponding signal, following the same workflow of Gainza et al. [20]. The final graph has 11966 nodes and 35892 edges.

**US election** Finally, we consider a social network dataset introduced by Jia and Benson [23], in which nodes represent United States counties and edges are estimated using the Facebook Social Connectedness Index. The target signal represents the county-wise outcome of the 2012 US presidential election, as values in the range $[-1, 1]$ (indicating the proportion of votes in

Table 2: Performance comparison ($R^2$) of typical INRs and generalised INRs (GINRs), using ReLU (r) or SIREN (s) activation.

|  | **INR (r)** | **INR (s)** | **GINR (r)** | **GINR (s)** |
|---|---|---|---|---|
| Bunny | 0.000 | 0.919 | **0.932** | 0.885 |
| Protein | 0.799 | 0.275 | **0.921** | 0.916 |

favour of one candidate *vs.* the other). The graph has 3106 nodes and 22574 edges.

**Results** In all cases, the generalised INR achieves an $R^2$ close to 1 indicating that the model can indeed learn non-trivial signals on non-Euclidean domains. We report the $R^2$ and mean squared error for all datasets in Tab. 1. We compare the true graph signals and those learned by the INR, for the bunny and protein, in Fig. 3. As a second test, we compare typical INRs trained on node coordinates with generalised INRs trained on the spectral embeddings. We consider the bunny and protein since they have input coordinates for the typical INRs. We also test different activation functions—ReLU or SIREN. We report in Tab. 2 the $R^2$ for a held-out set of nodes (to evaluate whether the INRs are overfitting instead of learning a meaningful representation).[4] All results are averaged over 5 runs and have negligible standard deviations. We see that the generalised INRs perform almost always better than the typical ones. Note that these experiments only verify that the model can indeed learn the target signals. We will test the transferability of the model in-depth in Sec. 4.2. In the following sections, we mostly focus on known spaces $\mathcal{T}$ or synthetic graphs so that we can better answer other research questions by having full control of the data.

**Size of the spectral embeddings** Typical INRs take as input the $d$-dimensional coordinates of points on the sampled lattice. In our general setting, the dimension $k$ of the spectral embeddings is a hyperparameter of the method. To evaluate the impact of $k$ on the performance of the INR, we train the same model for different values of $k$ on the Stanford bunny.

We report the results in Figure 4, showing that the model fails to learn a meaningful representation for $k \leq 3$. This is reasonable because at least 3 non-trivial eigenvectors are needed to correctly distinguish the main structural features of the bunny (ears, tail, etc.), as shown in Figure 5. We also see that, while the INR has no issues in learning the signal for $k = 100$, for lower values of $k$ the model struggles to represent the signal on the ears.

---

[1]Available at `https://graphics.stanford.edu/data/3Dscanrep/`

[2]$D_a = 0.64, D_b = 0.32, F = 0.06, K = 0.062$

[3]Protein Data Bank identifier: 1AA7

[4]We report training details in the supplementary material.

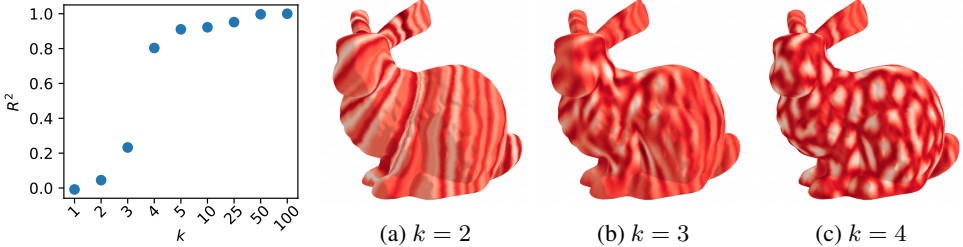

(a) $k = 2$      (b) $k = 3$      (c) $k = 4$

Figure 4: **Left:** $R^2$ *vs.* $k$; **Right:** signals learned by the INR for $k = 2, 3, 4$.

Figure 5 again provides a possible explanation for this result, since the first few eigenvectors tend to collapse on the ears and higher frequency eigenvectors are needed to distinguish points towards the narrow tips. We will show another consequence of this in Section 4.2. We also mention that the drop in performance for lower values of $k$ on the other two datasets is less pronounced than on the bunny, likely because of their lack of any high-frequency features.

### 4.2 Transferability of generalised INRs

Because the generalised spectral embeddings are a discrete approximation of the continuous eigenfunctions of the Laplace-Beltrami operator on $\mathcal{T}$, we can apply a trained INR to a different graph realisation of $\mathcal{T}$ as long as its spectral structure is consistent with the training data.

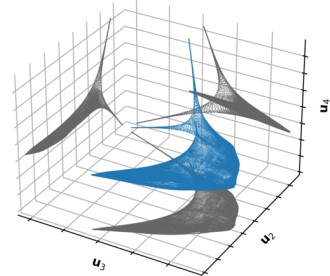

Figure 5: Spectral drawing (blue) of the bunny using the first three non-trivial eigenvectors. In grey, 2d projections of the 3d drawing.

In practical terms, we need to know 1) how to sample new nodes and 2) how the nodes are connected. If we know $\mathcal{T}$, then the problem simply boils down to choosing a sampling strategy for the points and a suitable measure of closeness (*e.g.* a kernel or edge-generating function like the k-nearest neighbours algorithm). If $\mathcal{T}$ is unknown, then we assume to observe a new graph through the same process that generated the training data (*e.g.*, we observe new social interactions among the same or a similar set of people).

Note that, when sampling a new graph from $\mathcal{T}$, the spectral embeddings will be a slightly different (possibly better) approximation of the continuous eigenfunctions. As this effect is more evident on high-frequency eigenvectors, we empirically observed that training the INR with a smaller $k$ and ReLU activations improved its transferability (*cf.* Table 2). See the supplementary material for more details. We investigate the transferability of generalised INRs in two settings.

**Transferring to similar graphs** As a first example, we study the case in which $\mathcal{T}$ is unknown and we only observe the discrete graphs. We consider a stochastic block model (SBM) with two communities of equal size (in total, $n = 1000$), parametrised by an inter-connection probability $p$ and an intra-connection probability $r$.

We train an INR on a graph with a strong community structure ($p = 0.1, r = 0.5$) and a graph signal indicating the community of each node (effectively a node classification task). Then, we test the INR on graphs sampled in the ranges $p \in [0.1, 1]$ and $r \in [0.1, 1]$ to test the robustness of the model to changes in the community structure.

We report our results in Figure 6. First, we note that perfectly recovering the correct clusters is only theoretically guaranteed in certain regions of the parameter space [1], delimited with an orange line in the figure.[5] We see that the trained INR transfers perfectly to all graphs sampled in this region and that its performance sharply drops only for $p > r$, where graphs have no community structure (see Figure 6 a, b). Additionally, we see that the INR maintains a good performance also in the region between the theoretical boundary and the line $p = r$, *i.e.*, a region in which graphs still exhibit some community structure although perfect recovery is not guaranteed. Overall, this indicates that training the INR on an instance of a graph allows us to transfer it to graphs with a similar structure.

---

[5]The region is above curve $r = \ln(n)/n \cdot \left( \sqrt{2} + \sqrt{np/\ln(n)} \right)^2$ [1].

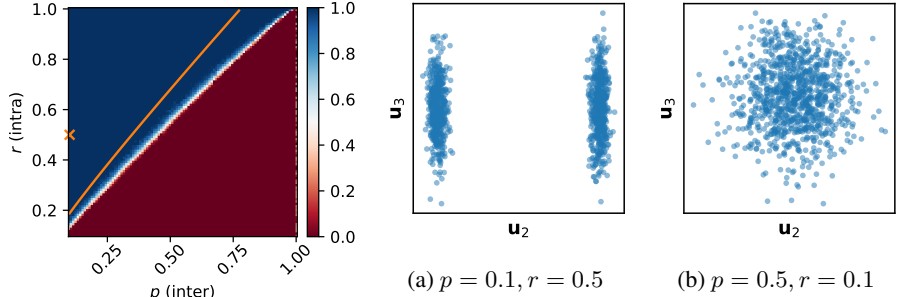

(a) $p = 0.1, r = 0.5$       (b) $p = 0.5, r = 0.1$

Figure 6: **Left:** normalised mutual information at test time, evaluating on graphs with different inter- and intra-connection probabilities. We trained the INR on $p = 0.1$ and $r = 0.5$, marked with $\times$ on the plot. The orange line indicates the theoretical boundary above which it is possible to perfectly recover the true signal. Best viewed in colour. **Right:** the first two non-trivial eigenvectors of two SBMs, respectively exhibiting strong and no community structure.

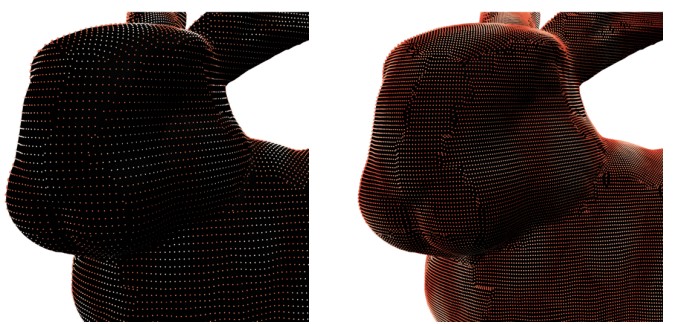

Figure 7: Zoomed-in comparison between the nodes of the training graph (left) and its super-resolved version (right), plotted on a black surface to aid visualisation. An interactive version in vector graphics of this figure is available in the supplementary material.

Figure 8: Distribution of the error in the super-resolution experiment.

**Super-resolution** As a second test case, we consider the super-resolution of a signal. We consider again the Stanford bunny mesh with the same texture graph signal as Section 4.1 and we apply Loop's method for mesh subdivision to obtain a higher-resolution mesh with 139122 nodes and 416929 edges [35]. Then, we compute the spectral embeddings for the new graph and predict the signal with the INR trained on the low-resolution graph.

Because the unit-norm eigenvectors are only unique up to sign, the INR will transfer to a different graph only if its eigenvectors are aligned to the training ones (*i.e.*, similar nodes must have similar spectral embeddings — in the previous experiment, we did not have to deal with this problem due to symmetry in the SBM). However, in many cases, a simple comparison of the eigenvectors' histograms provides a useful heuristic for automatically aligning them. For simplicity, here we also verify the alignment of the two eigenbases manually.

We show in Figure 7 a qualitative comparison between the training and super-resolved signals, from which we see that the INR can predict the graph signal correctly. Quantitatively, since mesh subdivision does not remove the original nodes from the mesh, we can compute the $R^2$ between the training signal and part of the super-resolved signal. The model achieves an $R^2$ of 0.39 compared to a training $R^2$ of 0.99. However, this apparent drop in performance is mostly due to the high-frequency ears of the bunny, for which the model evidently overfits and cannot tolerate even small changes in the eigenvectors. Figure 8 shows the distribution of the squared error between the ground truth and predicted signal, from which we see that the error is close to zero for most of the nodes (note that the y-axis is logarithmic). By ignoring the nodes above the 90th percentile of the squared error distribution, the $R^2$ score is 0.94. Increasing $k$ resulted in a better prediction of the ears region (*cf.* Figure 4) at the cost of worse overall transferability.

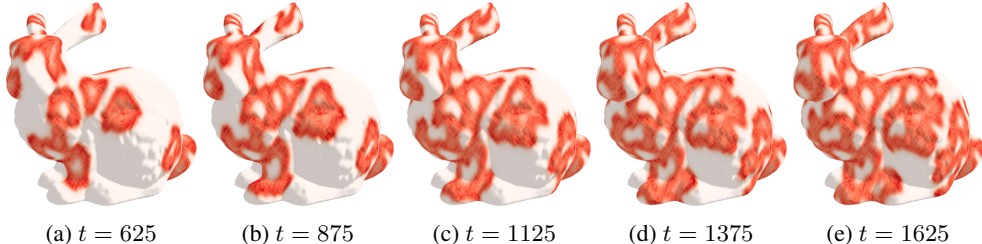

| (a) $t = 625$ | (b) $t = 875$ | (c) $t = 1125$ | (d) $t = 1375$ | (e) $t = 1625$ |

Figure 9: Signals predicted by the conditional INR $f_\theta(\mathbf{e}_i, t)$ at equispaced time steps $t \in [625, 1625]$. All time steps shown were not part of the training set. Also, each test time step is as far as possible from its two closest training samples (*i.e.*, training samples are at $t \equiv 0 \pmod{10}$ while test samples are at $t \equiv 5 \pmod{10}$). An animated version of this figure with 600 predicted samples is available in the supplementary material.

## 4.3 Conditional generalised INRs

We also consider the setting in which a generalised INR is conditioned on a latent vector $\mathbf{z} \in \mathbb{R}^q$ to allow the representation of different signals by the same neural network $f_\theta(\mathbf{e}_i, \mathbf{z})$. Unlike the Euclidean setting, here we can investigate two cases: 1) learning a conditional INR for different signals on the same domain (the typical setting), and 2) learning a conditional INR for different signals on different domains.

**Reaction-diffusion process**   For the first case, we train a generalised INR to approximate the Gray-Scott reaction-diffusion process that we used to generate the graph signal in Section 4.1, given by Equation (2). We consider a signal $f(v_i, t)$, where $t \in \mathbb{N}$ is the number of updates since the start of the simulation. We evolve the system for 3000 steps starting from the same random initialisation used in Section 4.1 and we train the INR $f_\theta(\mathbf{e}_i, t)$ using 300 time steps sampled at $t \equiv 0 \pmod{10}$. The graph is fixed throughout the evolution, so the INR must learn to exploit the conditioning input $t$ to correctly predict the signal at different times.

At test time, we predict the signal for all the remaining time steps and compare the prediction with the ground truth signals. The INR achieves a test $R^2$ of $0.999 \pm 0.006$ indicating that it successfully learned to approximate the true dynamics. We report in Figure 9 a few samples predicted by the INR at test time. By comparison, a simple baseline using linear interpolation between training samples achieves a similar $R^2$, while requiring to keep all 300 training steps in memory. The INR, on the other hand, represents the signal and its evolution with a single set of weights and a scalar control parameter.

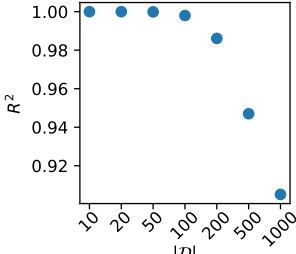

Figure 10: $R^2$ *vs.* size of the dataset $|\mathcal{D}|$.

**Multi-protein INR**   For the second case, we consider a dataset $\mathcal{D} = \{(\mathcal{T}_m, f_m)\}$ of protein surfaces $\mathcal{T}_m$ with associated electrostatic signals $f_m : \mathcal{T}_m \to \mathbb{R}$ for $m = 1, \dots, |\mathcal{D}|$, obtained following the same protocol as Section 4.1. We configure the neural network $f_\theta(\mathbf{e}_i, \mathbf{z}_m)$ as an autodecoder [41] where $\mathbf{z}_m \in \mathbb{R}^z$ is a vector of free parameters learned end-to-end alongside the weights of the neural network. We assign a separate vector $\mathbf{z}_m$ to each training sample, allowing the model to find the best latent representation for the dataset. Since the continuous surface and the associated signals are unique to each protein in the dataset, the latent vector $\mathbf{z}_m$ must capture both aspects. We set the latent vector size $z = 8$, which we empirically found to give consistently good performance across datasets. Larger values of $z$ did not significantly improve the performance.

We test the ability of the model to represent datasets of different sizes, by sampling random proteins from the dataset used in reference [20]. Figure 10 shows the $R^2$ of the model for different values of $|\mathcal{D}|$. Understandably, the performance decreases as the number of samples grows, although the model achieves $R^2 \geq 0.9$ for datasets of up to 500 different proteins. Overall, these results indicate that the model can represent many domains and signals with a single set of weights.

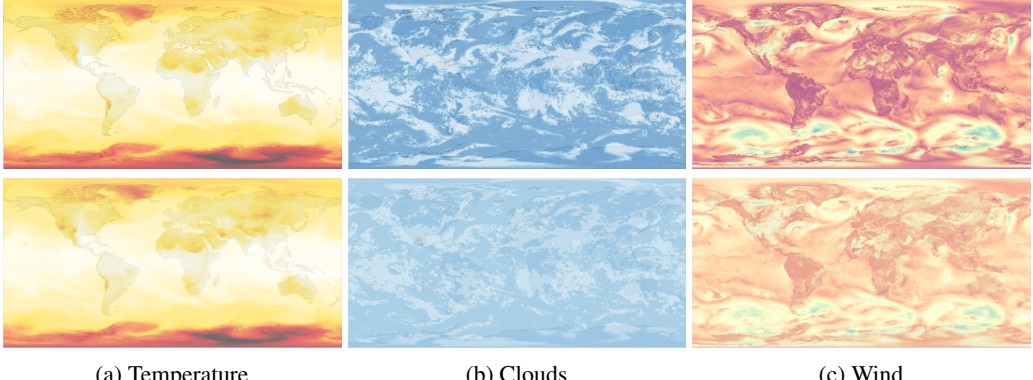

| (a) Temperature | (b) Clouds | (c) Wind |

Figure 11: **Top**: prediction of the INR for $t = 4.5$ (randomly chosen, not in the training data) at the original resolution. **Bottom**: The super-resolved versions of the top row. Note that the apparently lighter colours are simply artifacts of the plotting software. Best viewed in colour and high resolution. An animated and uncompressed version of this figure is available in the supplementary material.

### 4.4 Weather modelling

As a final experiment, we consolidate all results from previous sections to showcase an application of generalised INRs in a real-world setting. We consider signals $f(v_i, t)$ representing hourly meteorological measurements on the spherical domain of the Earth's surface. Our goal is to train a conditional INR at a given spatial and temporal resolution and then use it to super-resolve the signal over space and time.

**Data** We collected data from the National Oceanic and Atmospheric Administration (NOAA) Operational Model Archive and Distribution System, specifically from the Global Forecast System (GFS). The data consists of three different signals representing: the dew point temperature 2m above-ground measured in K (*dpt2m*), the atmosphere's total cloud cover percentage (*tcdcclm*), and the surface wind speed in m/s (*gustsfc*). We consider a 24-hour period sampled at 1-hour increments over an equiangular spherical grid sampled at 1° increments.[6] We generate a spherical mesh from the 65160 locations of the grid by computing their convex hull with the Qhull software [4], for a total of 193320 edges.

**Results** We train a conditional INR $f_\theta(\mathbf{e}_i, t)$ at the given spatial and temporal resolution, for $t = 0, \dots, 23$. Then, we evaluate its predictions on a high-resolution spherical mesh with 258480 nodes and 773280 edges (obtained with the same procedure of Section 4.2), and at time increments of 30 minutes. We report in Figure 11 the prediction of the INR for a random test time, on both the original mesh and the high-resolution one. We refer the reader to the animated version of the figure available in the supplementary material for a better appreciation of the results. Overall, our results confirm that generalised INRs can model complex real-world signals on non-Euclidean domains, learning a realistic continuous approximation of the signals from low-resolution samples.

## 5 Conclusion

We have presented the problem of learning generalised implicit neural representations for signals on non-Euclidean domains. Our method learns to map a spectral embedding of the domain to the value of the signal, without relying on a choice of coordinate system. We have shown applications of our method on biological, social, and meteorological data, highlighting the potential usefulness of such INRs in a wide range of scientific fields. We hope that future work will explore more real-world applications of our method.

---

[6]Specifically the period from 00:00 to 23:59 of May 9, 2022. All times $t$ given in the text are in hours relative to the start of the period.

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
