# Supplementary material for "Generalised Implicit Neural Representations"

**Daniele Grattarola**
EPFL
Lausanne, Switzerland
daniele.grattarola@epfl.ch

**Pierre Vandergheynst**
EPFL
Lausanne, Switzerland
pierre.vandergheynst@epfl.ch

## A    Solving the Poisson equation with generalised INRs

One interesting application of INRs is to train them using the derivatives of the target signal as supervision. This idea, which was introduced by Sitzmann et al. [1], can also be applied to the generalised case.

Specifically, in this experiment we consider the bunny reaction-diffusion texture and train a generalised INR to minimise:

$$\mathcal{L} = \|\mathbf{Lf} - \mathbf{Lf}_\theta\|, \tag{1}$$

where $\mathbf{L}$ is the graph Laplacian, $\mathbf{f} = [f(v_1), \ldots, f(v_n)]^\top \in \mathbb{R}^n$ is the target graph signal, and $\mathbf{f}_\theta = [f_\theta(\mathbf{e}_1), \ldots, f_\theta(\mathbf{e}_n)]^\top \in \mathbb{R}^n$ is the signal predicted by the INR.

We report in Figure 1 the original texture, its Laplacian, and the reconstructed signal, as predicted by the INR. We see that the model is able to correctly reconstruct the signal, although with some oversmoothing.

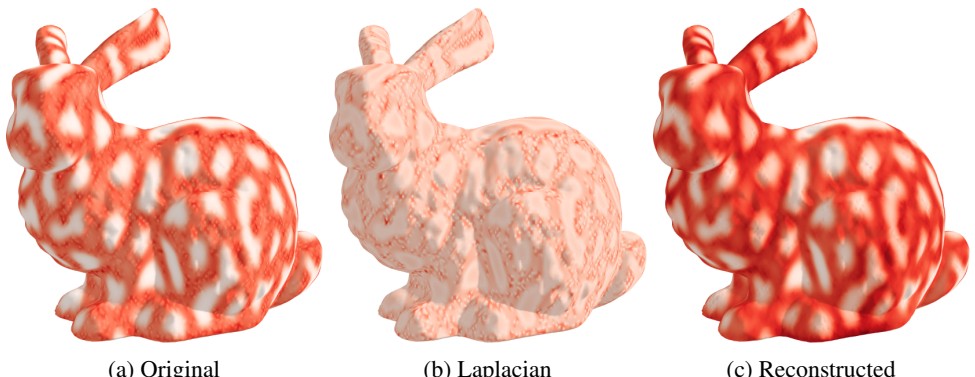

(a) Original             (b) Laplacian             (c) Reconstructed

Figure 1: The original signal, its laplacian, and its reconstructed version for the bunny reaction-diffusion texture.

## B    Changes in the experimental setting

**Spectral embeddings**    In the transferability experiments, we observed that changes in the high-frequency eigenvectors of different graph realizations of $\mathcal{T}$ caused the INR to perform poorly at test time.

For this reason, we used smaller spectral embeddings of size $k = 3$ for the SBM experiment (which was enough to highlight the community structure of the graphs) and $k = 7$ for the super-resolution

36th Conference on Neural Information Processing Systems (NeurIPS 2022).

experiment with the bunny (which we found by trial and error balancing the speed of convergence and the final performance).

In the weather experiment, the equiangular grid returned by GFS has a higher point density at the poles. This non-uniformity has the effect that the first 66 eigenvectors are constant everywhere except at a very concentrated region around the poles, and they tend to change a lot when sampling more points (making it harder to transfer the trained INR to the high-resolution graph). To improve stability, we removed the first 66 almost-trivial eigenvectors and trained the INR only on the remaining 34.

**Architecture** We also observed that the SIREN multi-layer perceptron was sometimes too sensitive to small changes in the spectral embeddings. For this reason, we searched for a different architecture that would make the INR more transferable to different graphs.

The alternative architecture has the following characteristics:

- ReLU activations;
- Uniformly distributed initial weights;
- 8 layers instead of 6 (we searched for the best value in $[4, 10]$);
- Learning rate of $10^{-3}$ (the default of $10^{-4}$ was not enough for the model to converge);

Note that we still kept the SIREN architecture for all other experiments as it exhibited better speed of convergence and lowest training loss compared to the alternative architecture.

## C  Training details

In the second experiment of Section 4.1, we train the models using the default setting. We create random training, validation and test splits with a proportion of 80%, 10%, and 10% of the node set. We train the models to convergence, interrupting training if the validation loss does not improve for 1000 steps (we also lower the learning rate annealing to have a patience of 500 steps). To compare the different activations, we simply swap the activation function and initialisation schemes, leaving everything else the same.

## D  Alternative normalisation of the Laplacian

Using different Laplacian normalisation schemes for computing the spectral embeddings did not yield significant differences in performance when training generalised INRs.

While the magnitude of individual embeddings will change across normalisations, they are qualitative similar and provide a reasonable encoding for a node's position. For visualizing this equivalence, we report in Figure 2 eigenvectors $\mathbf{u}_k$ for $k = [1, 5, 10, 20, 50]$ for three types of Laplacian: the combinatorial Laplacian, $\mathbf{L} = \mathbf{D} - \mathbf{A}$; the symmetrically normalised Laplacian, $\mathbf{L}_n = \mathbf{D}^{-1/2}\mathbf{L}\mathbf{D}^{-1/2}$; and the random walk normalised Laplacian $\mathbf{L}_{rw} = \mathbf{D}^{-1}\mathbf{L}$.

## References

[1] Vincent Sitzmann, Julien Martel, Alexander Bergman, David Lindell, and Gordon Wetzstein. Implicit neural representations with periodic activation functions. *Advances in Neural Information Processing Systems*, 33:7462–7473, 2020. 1

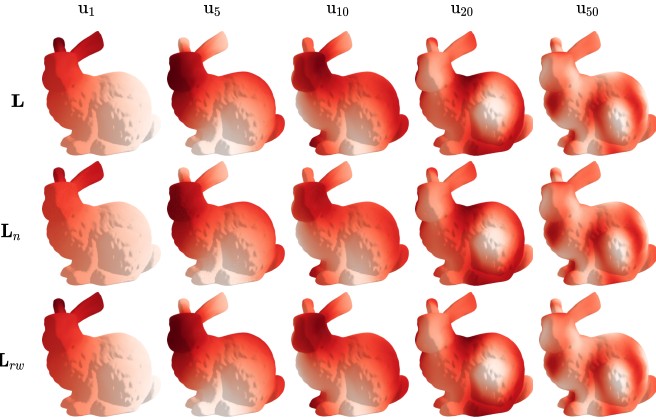

Figure 2: Eigenvectors $\mathbf{u}_k$ for $k = [1, 5, 10, 20, 50]$ for different Laplacian normalisation schemes, plotted on the Bunny mesh. Colours indicate intensity and colour scales are not shared between different rows.