# OpenReview forum: "Generalised Implicit Neural Representations"
_NeurIPS.cc/2022/Conference — NeurIPS 2022 Accept_

### Official Review · Reviewer_8jka · 2022-07-03

**Rating:** 7
**Confidence:** 4
**Soundness:** 3 good
**Presentation:** 3 good
**Contribution:** 3 good

**Summary:**

The authors propose to parameterize implicit neural representations (INRs) as neural networks that take as input the Laplacian eigenvectors of a non-Euclidean domain. This allows for learning INRs over spaces without an external coordinate system. These INRs are tested over various application areas, each of which defines a suitable discrete graph for computing the Laplacian eigenvectors from.

--------

After reading the authors replies, I have increased my score to a 7. The comparison with other choices / baselines, and the improved exposition / comparison to literature have greatly improved the paper in my eyes.

**Questions:**

1. For the Bunny mesh, do you use the cotangent Laplacian scheme, or do you still use the combinatorial graph Laplacian of the mesh? I'm curious as the cotangent Laplacian scheme is widely used for meshes, but there appears to be no mention of specific Laplacian normalization for this mesh.
2. You treat the Earth as a sphere, correct? If so, it would be possible to evaluate the exact Laplacian eigenfunctions, how does this compare?

**Limitations:**

Are the authors aware of [Koestler et al. 2022]? These two papers do the same thing. Given that the idea is so simple, this would be an issue for novelty. I see that it is from about 2 months before the NeurIPS submission deadline, so we may ignore the effect on novelty it if the policy requires.

References
[Koestler et al. 2022] Intrinsic Neural Fields: Learning Functions on Manifolds. https://arxiv.org/abs/2203.07967

**Strengths And Weaknesses:**

Strengths:
1. The paper is simple and straightforward, in a good way.
2. This is a natural generalization of the Euclidean case, in which signals are usually defined on a grid. INRs have found much success in the Euclidean case, and they can presumably be very useful for non-Euclidean domains.

Weaknesses:
1. Please be more precise with the claim that the INRs are ``equivariant under the symmetry group of the domain.'' In particular, it would be great to include a reference, or to include a precise statement and proof. I think what you mean is that the eigenvectors are equivariant under the symmetry group, and / or the entire INR algorithm (i.e. the mapping from input to final trained model) is equivariant.
2. Would be nice to compare different choices of Laplacian normalizations, or to compare against using the ground truth eigenfunctions of a continuous surface like the sphere (see Questions section for more). E.g. various normalized graph Laplacians are often used for many applications.
3. The ad-hoc manual choice of sign for the eigenvectors seems a bit hacky for the super-resolution task. Past work has given ways to get rid of sign ambiguity (there is also a basis ambiguity for higher dimensional eigenspaces), see [Lim et al. 2022], which also tests on a generalized INR task on non-Euclidean data.

Minor notes:
1. For graph positional encodings, in addition to random walk PEs there are various other PEs that are sign invariant, see [Lim et al. 2022] Proposition 3 for examples. These include the well known Heat Kernel Signature [Sun et al. 2009].
2. May be worth pointing out that the sinusoidal positional encodings / Fourier features used in Euclidean INRs are eigenfunctions of the Laplace operator on Euclidean space. Thus, your method is a direct generalization.

References:
[Lim et al. 2022] Sign and Basis Invariant Networks for Spectral Graph Representation Learning. https://arxiv.org/abs/2202.13013
[Sun et al. 2009] A Concise and Provably Informative Multi-Scale Signature Based on Heat Diffusion. Eurographics Symposium on Geometry Processing.

---

> ### Author Response · Authors · 2022-08-02
> **Reply to reviewer 8jka (part 1)**
>
> We thank the reviewer for the time spent reviewing our paper and for helping us improve our work.
>
> We have addressed all comments and suggestions made by the reviewer, adding new discussions in the paper as needed.
>
> We address the reviewer's concerns below.
>
> **Weaknesses**
>
> > Please be more precise with the claim that the INRs are "equivariant under the symmetry group of the domain."
>
> We realised that our discussion on equivariance was only meaningful when considering a coordinate system to identify the position of the nodes.
> In our case, however, we do not rely on the existence of a coordinate system and instead represent each node using the spectral embeddings, which are an intrinsic property of the domain.
>
> We therefore have changed our claim to only say that, being the spectral embeddings an intrinsic property of the domain, the INR will be independent of any possible choice of coordinate system.
> We believe this observation to be more precise and useful in practice (in the end, a user might mostly care that they don't have to align their point clouds in order to train a generalised INR).
>
> > Would be nice to compare different choices of Laplacian normalizations.
>
> We thank the reviewer for the suggestion.
>
> We have tried different Laplacians including the combinatorial, normalised, and random walk normalised Laplacians. We did not obtain significantly different results in our experiments.
>
> We have now added a section in the supplementary material showing that there are no substantial qualitative differences between the embeddings obtained from the three types of Laplacian (all provide reasonable positional encodings, despite minor differences).
>
> > The ad-hoc manual choice of sign for the eigenvectors seems a bit hacky for the super-resolution task. Past work has given ways to get rid of sign ambiguity (there is also a basis ambiguity [...]), see [Lim et al. 2022]
>
> We agree that manual alignment is sub-optimal. However, the heuristic mentioned on lines 264-266 worked reliably and our manual intervention in Section 4.2 was only to double-check the alignment (we have changed the incorrect wording in the text).
>
> We have added the reference suggested by the reviewer as a possible way to address the sign and basis ambiguity.
>
> **Minor notes**
>
> > For graph positional encodings, [...] there are various other PEs that are sign invariant, see [Lim et al. 2022] Proposition 3 for examples. These include the well known Heat Kernel Signature [Sun et al. 2009].
>
> We thank the reviewer for pointing out these works. We have included all these references in the "Alternative approaches" section.
>
> > May be worth pointing out that the sinusoidal positional encodings / Fourier features used in Euclidean INRs are eigenfunctions of the Laplace operator on Euclidean space.
>
> We thank the reviewer for this suggestion. We have added this consideration to the "Related works" section (where we talk about positional encodings).
>
> **Questions**
>
> > For the Bunny mesh, do you use the cotangent Laplacian scheme, or do you still use the combinatorial graph Laplacian of the mesh?
>
> We use the combinatorial Laplacian in all experiments.
>
> As commented above, other choices of Laplacian are possible (e.g., when designing an INR specifically for point clouds, one might use the cotangent Laplacian) but we leave this exploration to future research.
>
> > It would be possible to evaluate the exact Laplacian eigenfunctions [on the sphere], how does this compare?
>
> Using the analytic eigenfunctions of the sphere does not significantly change the results. This is because the eigenfunctions are already well-approximated by the eigenvectors, given that we have a large number of nodes.
>
> The main difference is that using eigenfunctions allows us to query arbitrary points on the sphere without remeshing.
> However, in the general case that we are addressing in the paper, we do not know the domain or the coordinates and therefore we would not know how to compute the eigenfunctions.
>
> We have added a discussion about this in Section 3 of the paper.

---

> > ### Author Response · Authors · 2022-08-02
> > **Reply to reviewer 8jka (part 2)**
> >
> > **Limitations**
> >
> > > Are the authors aware of [Koestler et al. 2022]?
> >
> > We became aware of this work after submitting our paper.
> > We have added a reference to the paper, as well as a discussion on the differences.
> >
> > There are important differences between the two works:
> >
> > 1. We focus on a wider range of data and experiments, including transferring to completely different graphs (SBM), conditioning the INR, and modelling dynamical systems. In the revised version of our paper, we have also added an experiment about solving the Poisson equation in the generalised case.
> > 2. We propose a different scheme for computing generalised embeddings that does not rely on knowing node coordinates (Koestler et al. use the _robust Laplacian_, which is specifically designed for point clouds).
> > 	This is an important difference that allows us to train and transfer INRs on arbitrary graphs.
> > 1. Koestler et al. focus their analysis on NTK theory, a direction that is orthogonal to what we have explored in our work.
> >
> > We believe that there are no concerns for novelty, since the two works are comparable but complementary, and that both papers can be useful to the INR community.

---

> > > ### Comment · Reviewer_8jka · 2022-08-02
> > > **Reply to the authors**
> > >
> > > We thank the authors for their quick and thorough reply. The clarification about intrinsic symmetries, comparison to related work, sign choice in super-resolution experiments, and comparison to other baselines (e.g. Euclidean INR, other normalizations, Koestler et al, using exact continuous eigenfunctions) are all (in my opinion) very beneficial to the work, and make it a more complete work. Also, the discussion with other reviewers bring up several interesting points. I find this work to be useful in terms of an exploration of the various things that one can do with INRs defined on estimated Laplacian eigenfunctions, and also as a reference for various considerations that may be useful for using such INRs. Thus, I will raise my score to a 7.

---

### Official Review · Reviewer_NW5a · 2022-07-09

**Rating:** 6
**Confidence:** 3
**Soundness:** 3 good
**Presentation:** 4 excellent
**Contribution:** 3 good

**Summary:**

The paper proposes to extend implicit neural representation (INR) to non-euclidean settings, in particular graphs. This is achieved by using the top K eigenvectors of the graph laplacian to replace euclidean coordinates in INR. Experiments on several datasets, including protein and social networks, demonstrate the ability of the generalized INR to represent the signal and generalize to similar graphs faithfully.


**Questions:**

- The paper claims that the proposed model is “equivariant” to domain symmetries. I understand that the generation process is “invariant” to the symmetry transformations that leave the graph invariant. What am I missing? Also, these transformations are not necessarily the symmetries of the original manifold but those of the graph.

- For experiments on conditional generation with the Stanford bunny, the figure (figure 9) does not convey the error in the generation process. The same is true for weather modelling (figure 10). Since you have the ground truth, please report the error in your approach. You can also compare to a simple baseline that, for example, uses averaging for interpolation?


**Limitations:**

Limitations are not currently discussed.

**Strengths And Weaknesses:**

**Strengths**

- The paper has a polished and easy-to-read presentation, with many figures that help present the main idea and experimental results.
- The paper does a good job of reviewing related works both on INR and related developments in geometric deep learning.
- A wide range of experiments considering different tasks and families of graphs are used for evaluation.

**Weaknesses**

- Some experiments lack proper baselines and quantitative evaluation. Also, although quite pleasing to the eye, figures do not convey the quality of results since comparison to ground truth is missing.
- Discussions and experiments put more emphasis on reconstruction rather than generalization in using INR, especially generalization to new graphs where this approach can be most interesting could receive more attention.

---

> ### Author Response · Authors · 2022-08-02
> **Reply to reviewer NW5a**
>
> We thank the reviewer for the time spent reviewing our paper and for the overall positive feedback on our work.
>
> We have addressed all comments made by the reviewer, and we feel that the paper has improved as a consequence.
>
> **Weaknesses**
>
> > Some experiments lack proper baselines and quantitative evaluation. Also, although quite pleasing to the eye, figures do not convey the quality of results since comparison to ground truth is missing.
>
> We have added new baselines and additional quantitative results to the revised paper.
> Specifically, we report the test performance on a held-out set of nodes and a comparison with INR baselines trained on node coordinates instead of generalised embeddings (see Section 4.1).
>
> In most cases, the predicted and ground truth signals are indistinguishable by eye. For example, see the comparison between the predicted and ground truth signals in Figure 3. Adding such figures for each experiment would not convey much information.
>
> We do however report quantitative results whenever appropriate (e.g., in Tables 1 and 2; Figures 5, 6, 8 and 10; and the main text in Section 4.3).
>
> > Discussions and experiments put more emphasis on reconstruction rather than generalization in using INR, especially generalization to new graphs where this approach can be most interesting could receive more attention.
>
> We prioritised exploring different types of experiment to give a better sense of the possibilities that our method unlocks.
>
> However, Section 4.2 is entirely focused on generalisation to different graphs, and part of Section 4.4 is as well.
>
> We have also added a new experiment in Section 4.1 to show the generalisation of a trained INR on a held-out set of nodes.
>
> We are hopeful that future work will explore other practical applications of our idea.
>
> **Questions**
>
> > The paper claims that the proposed model is “equivariant” to domain symmetries. I understand that the generation process is “invariant” to the symmetry transformations that leave the graph invariant. What am I missing?
>
> We realised that our discussion on equivariance was only meaningful when considering a coordinate system to identify the position of the nodes.
> In our case, however, we do not rely on the existence of a coordinate system and instead represent each node using the spectral embeddings, which are an intrinsic property of the domain.
>
> Therefore, we have changed our claim to only say that, being the spectral embeddings an intrinsic property of the domain, the INR will be independent of any possible choice of coordinate system.
>
> We believe this observation to be more precise and useful in practice (in the end, a user might mostly care that they don't have to align their point clouds in order to train a generalised INR).
>
> > For experiments on conditional generation with the Stanford bunny, the figure (figure 9) does not convey the error in the generation process. The same is true for weather modelling (figure 10). Since you have the ground truth, please report the error in your approach.
>
> We report the test $R^2$ for the reaction-diffusion experiment on line 293.
> For the weather experiment, we do not have the ground truth because the data is sparse over time. However, we include a high-resolution and animated version of the figure in the supplementary material to better convey the quality of the learned signal.
>
> > You can also compare to a simple baseline that, for example, uses averaging for interpolation?
>
> We have implemented the baseline suggested by the reviewer and we now include the results and a discussion in the paper.
> When evaluating on the test set, the interpolation baseline achieves a similar $R^2$ to the INR, which is not surprising given that the dynamical system is well described by linear interpolation.
>
> We recall that the purpose of an INR is not to improve the performance of interpolation tasks, but to provide a different representation for a signal.
> While the interpolation baseline requires storing all 300 training time steps, the INR captures a compact global model of the signal using a relatively small set of weights.
>
> Additionally, INRs are fully parametric and differentiable, which allows one to integrate them into various optimization pipelines (e.g., see [1]).
>
> [1] Remelli et al. "Meshsdf: Differentiable iso-surface extraction." _Neural Information Processing Systems_ (2020).
>
> **Limitations**
>
> We have improved the discussion on the limitation of our approach in Section 3 and we also comment on possible alternative approaches and solutions to some of the issues that the approach has.

---

### Official Review · Reviewer_ofDT · 2022-07-10

**Rating:** 6
**Confidence:** 5
**Soundness:** 3 good
**Presentation:** 3 good
**Contribution:** 3 good

**Summary:**

This paper generalizes implicit neural representation (INR) to arbitrary topological spaces beyond the regular Euclidean space. Technically, the authors utilize a graph as an approximation of topological space, and compute its spectral embedding as the point-wise encoding to regress an INR. In their experiments, they show the learned INR can be transferable to other graph approximations of the topological space and their techniques are also applicable to conditional INRs.

**Questions:**

- Solving spectral embedding suffers from the problem of scalability, stability, multiplicity and ambiguity of sign. How did authors solve these practical challenges in this paper?

- Typical INR maps spatial coordinates to the corresponding value. How to bind generalized INR with positional information?

- Can generalized INR solve differential/integral equations like SIREN?


**Limitations:**

Generalizing INR to arbitrary topology seems a promising direction. However, the strategy of  approximation via graphs seems to have some limitations. In vanilla INR, arbitrary coordinates in Euclidean space can be directly fed into the INR and query for the value. However, graph discretizes the signal and thus only vertices in the graph can be fed into the INR. That being said, the represented signals are no more continuous. Authors may discuss this limitation and provide preliminary solutions to it.

**Strengths And Weaknesses:**

Strengths:

+ The paper is clearly written and easy to follow.

+ The authors put forward an interesting question, i.e., how to learn an INR over the domain of arbitrary topology. They further propose a simple yet effective and well motivated method. It also has lots of practical usage in geometric, molecular, and climate data representation.

+ The experimental discussion is comprehensive. It demonstrates different applications and verifies the transferability of the learned INR across graphs sampled from the same topological space.

Weaknesses:

- The reviewer cannot agree that the represented signals are still continuous. Approximating a topological space via a graph discretizes the signal space, which makes the input points restricted to the vertex set of the graph. To show the interpolation ability, the authors may first construct an over-dense graph, and then only train the INR on a subset of vertices.

- In regards to the experiment, there is no comparison with other methods. To demonstrate the benefit of involving topological information, authors may compare their model with Euclidean domain INR. For example, also show the texture map regression results on SIREN.

---

> ### Author Response · Authors · 2022-08-02
> **Reply to reviewer ofDT (part 1)**
>
> We thank the reviewer for the time spent reviewing our paper and for the overall positive feedback on our work.
>
> We have added new baselines and quantitative results, as well as a completely new experiment on solving differential equations, as suggested by the reviewer.
>
> We address the reviewer's comments below.
>
> **Weaknesses**
>
> > The reviewer cannot agree that the represented signals are still continuous. Approximating a topological space via a graph discretizes the signal space, which makes the input points restricted to the vertex set of the graph.
>
> The sampled graph signal is indeed discrete, in the same way that a grid of pixels is discrete. However, INRs are continuous functions defined on the entire space $\mathcal{M}$. As the number of nodes goes to infinity, the discrete graph converges to $\mathcal{M}$ and its eigenvectors converge to the continuous eigenfunctions (see Sections 2 and 3). This is no different than the Euclidean case.
>
> An important difference between generalised INRs and typical INRs is that we generally don't know the topological space that is discretised by the graph. For this reason, we must observe the graph in order to compute the inputs.
>
> This is necessary because, while in the Euclidean case we fully know the domain of the signal _a priori_, here we need to estimate the topology of the domain.
> We have added this important discussion to the "Method" section of the paper.
>
> > To show the interpolation ability, the authors may first construct an over-dense graph, and then only train the INR on a subset of vertices.
>
> We have added an experiment where we train an INR on a subset of vertices and test it on the remaining nodes.
>
> We report the results in Table 2, showing that the generalised INR achieves an average test $R^2$ of 0.913.
>
> We thank the reviewer for the suggestion.
>
> > To demonstrate the benefit of involving topological information, authors may compare their model with Euclidean domain INR. For example, also show the texture map regression results on SIREN.
>
> We have added a comparison between our generalised INR and two baselines trained on node coordinates (using SIREN and ReLU activations).
>
> We show that the generalised INR achieves much better performance than its counterparts, which in some cases are entirely unable to learn a correct representation.
>
> We thank the reviewer for the suggestion.
>
> **Questions**
>
> > Solving spectral embedding suffers from the problem of scalability, stability, multiplicity and ambiguity of sign. How did authors solve these practical challenges?
>
> We addressed the issues as follows:
>
> - Scalability: in practice, all our experiments were managed on a typical laptop. We comment on the cost of the method and on possible alternative ways to compute the positional encodings in Section 3 ("Complexity" and "Alternative approaches").
> - Stability: we did not encounter stability issues (the standard Scipy implementation of the eigendecomposition proved to be enough). Estimating only the low-frequency eigenvectors probably helped in this regard.
> - Multiplicity: we did not encounter this issue, although we realise that it is an important aspect to consider. We have now added a discussion where we address eigenbasis ambiguity and point to possible solutions.
> - Ambiguity of sign: we devised a simple heuristic based on the eigenvectors histogram that allowed us to reliably choose the correct sign for new graphs. The code for the heuristic is available in the supplementary material. We have also extended the discussion about sign ambiguity in the paper, proposing alternative solutions.
>
> > Typical INR maps spatial coordinates to the corresponding value. How to bind generalized INR with positional information?
>
> The graph positional encodings that we use in the paper carry that exact information, in a general way.
>
> Using pure coordinates is in fact a limitation of Euclidean INRs (as we also show in the new experiment), and the literature has moved past them in favour of sinusoidal activations or random Fourier features. In our case, the generalised spectral embeddings are a strict generalisation of this idea.
>
> > Can generalized INR solve differential/integral equations like SIREN?
>
> Yes, there is nothing that prevents our model to be supervised by the derivatives of the target signal.
>
> Following the reviewer's comment, we have added an experiment in the supplementary material showing a setting similar to the SIREN paper.
>
> We train a generalised INR using only the Laplacian of the target signal as supervision, and report the successfully reconstructed signal.

---

> > ### Author Response · Authors · 2022-08-02
> > **Reply to reviewer ofDT (part 2)**
> >
> > **Limitations**
> >
> > > In vanilla INR, arbitrary coordinates in Euclidean space can be directly fed into the INR and query for the value.
> >
> > We discuss the issue of querying the signal at arbitrary coordinates in Section 3: we propose to use the Nyström method as a fast way to estimate the generalised spectral embeddings when observing new nodes, similar to what is possible in the Euclidean case.
> >
> > This does not change the fact that one must estimate the local topology to predict the signal at a new location, but indicates a possible future direction for research.

---

> ### Comment · Reviewer_ofDT · 2022-08-09
> **Interesting work. Advocate acceptance**
>
> Thanks for reply and sorry for the late response. Authors' reply has resolved all my questions. I'll keep my score and advocate acceptance.

---

### Author Response · Authors · 2022-08-02
**Comment to reviewers and chairs**

We are glad that all reviewers agree that our paper is interesting, useful for the community, and thorough.

We sincerely thank the reviewers for the time taken to review our paper.

We have addressed all comments made by the reviewers, and we feel that the paper has improved as a consequence. We have uploaded a revision.

Here is a brief summary of the most important changes:

1. We have added new baselines and quantitative results.
2. We have added two new experiments (training on a subset of nodes in Section 4.1 and solving the Poisson equation in Appendix A).
3. We have simplified the discussion about equivariance in order to be more precise.
4. We have added some missing references and a discussion about some relevant prior work.
5. We have added some interesting discussions that emerged from the reviewers' comments.
6. We have extended the discussion on the limitations of our method.

We remain available to clarify any other doubts.

---

### Meta-Review · Area_Chair_hLZz · 2022-08-26

**Recommendation:** Accept
**Confidence:** Less certain

**Metareview:**

There is a clear consensus amongst the reviewers that the results are interesting, the manuscript is easy to read, and would be interesting to the NeurIPS community.  One of the main criticisms was the lack of baselines to compare the authors results against, and this was resolved by the authors in revision which encouraged one reviewer to advance their recommendation from weak accept to accept; this can be expected to also have been appreciated by other reviewers.

**Award:**

No

---

### Decision · Program_Chairs · 2022-09-14

Accept